# Investigation of the Microstructure and Wear Properties of Conventional Laser Cladding and Ultra-High-Speed Laser Cladding Alloy Coatings for Wheel Materials

Qian Xiao [1,2], Jinlong Xia [1,2], Xueshan Gao [1,2], Wenbin Yang [1,2,3,*], Daoyun Chen [1,2], Haohao Ding [4] and Yao Wang [2]

[1] Key Laboratory of Railway Industry on Intelligent Operation and Maintenance for Locomotive and Vehicle, East China Jiaotong University, Nanchang 330013, China; jxralph@foxmail.com (Q.X.); xia1260662404@163.com (J.X.); gao.xueshan@foxmail.com (X.G.); chendaoyun@ecjtu.edu.cn (D.C.)

[2] Key Laboratory of Conveyance Equipment, Ministry of Education, East China Jiaotong University, Nanchang 330013, China; wy_jxnc_ecjtu@126.com

[3] Key Laboratory of Urban Rail Transit Intelligent Operation and Maintenance Technology & Equipment of Zhejiang Province, Zhejiang Normal University, Jinhua 321005, China

[4] Tribology Research Institute, State Key Laboratory of Traction Power, Southwest Jiaotong University, Chengdu 610031, China; haohao.ding@swjtu.edu.cn

[*] Correspondence: yangwenbin@ecjtu.edu.cn

**Abstract:** In this paper, Fe-based and Co-based alloy powders were chosen to perform laser cladding on wheel materials through conventional laser cladding (CLC) and ultra-high-speed laser cladding (UHSLC) processes, respectively. The microstructures, element distribution, phase composition and hardness of the Fe-based alloy and Co-based alloy coating layers using the CLC and UHSLC processes were compared and analysed. The results show that the CLC and UHSLC alloy coatings were dense and free of defects such as pores and cracks. Compared with the CLC alloy coating, the grain size of the UHSLC alloy coating was smaller, the coating composition was close to the powder design composition, and the distribution of Cr within and between the grains was more uniform. The Fe-based coating was mainly composed of (Fe, Ni) and $Cr_7C_3$, and the Co-based coating was mainly composed of $\gamma$-Co and $Cr_{23}C_6$. It was found that the cooling rate of the CLC alloy coating was smaller than that of the USHLC, and the hardness of the CLC alloy coating was less than that of the USHLC. The average hardness of the UHSLC Fe-based and Co-based alloy coatings was 709 HV and 525 HV, respectively. The average hardness of the CLC Fe-based and Co-based alloy coatings was 615 HV and 493 HV, respectively. The rolling friction and wear tests were carried out with the CLC-treated and UHSLC-treated wheel specimens on the GPM-30 rolling contact fatigue testing machine. The results showed that the wear rate of the UHSLC alloy coating on the wheel specimens was significantly lower than that of the CLC alloy coating on the wheel specimens. The wear rates of the UHSLC Fe-based and Co-based alloy coatings on the wheel specimens were reduced by 40.7% and 73.8%, respectively. It was demonstrated that the wear resistance of the USHLC alloy coatings was better than those of the CLC alloy coatings. The CLC alloy coating exhibited more severe fatigue damage with small cracks. Furthermore, the damage of the UHSLC alloy coating was relatively minor, with slight spalling. The Co-based alloy coating exhibited superior wear properties with the same laser cladding process.

**Keywords:** conventional laser cladding; ultra-high-speed laser cladding; wheel materials; wear; damage

## 1. Introduction

As a key component of a train, the condition of the train wheels directly affects the train operation safety. Complex railroad line conditions and the increase in train speed and

axle weight may cause serious wear damage and rolling contact fatigue (RCF) damage to the train wheels [1–3], which affects the reliability of train operation. At present, defects are mainly removed by turning and restoring the tread profile, which leads to substantial wheel material waste and significantly shortens wheel life. The advantages of laser cladding technology for repairing local damage to train wheels are obvious, which can effectively improve the wear properties [4] of wheel materials, thus enhancing the service life of wheel materials and the reliability and comfort of trains during operation.

Conventional laser cladding (CLC) technology can achieve surface modification and damage repair, but its industrial application is limited by its low efficiency and high dilution ratio. In contrast, ultra-high-speed laser cladding (UHSLC) technology has higher scanning speeds and lower dilution rates, making it an emerging surface modification technology that can better meet the needs of industrial applications. Furthermore, the UHSLC process has the advantages of high bond strength [5–7] and a small heat-affected zone (HAZ), which is expected to become an important technology in the field of surface modification and surface repair.

Currently, many scholars apply laser cladding technology in the field of surface modification and damage repair of wheel/rail materials. For example, Lewis et al. [8] used an eddy current crack detector to monitor fatigue cracks in wheel materials, and the results showed that the coating had excellent wear resistance properties. Zhu et al. [3] prepared three stainless-steel alloy coatings on the wheel material separately and compared the wear and RCF performance of the specimens. The results showed that the coating had higher wear resistance and resistance to RCF, far exceeding the substrate. Li et al. [9] and Shen et al. [10] prepared an AISI 431 alloy coating via the UHSLC process. The effect of laser cladding speed on the microstructure and corrosion resistance of the alloy coating was discussed and compared with the CLC alloy coating. It was concluded that the faster the cladding speed is, the more uniform the distribution of elements in the alloy coating, and the uniform distribution of Cr within the tissue is beneficial for enhancing the corrosion resistance of the coating. Wang et al. [11] selected a Co-based alloy powder to repair the wheel/rail roll surface, and the prepared Co-based alloy coating was subjected to heavy-duty experiments. The results showed that the hardness of the coating gradually decreased as the coating depth increased and approached the hardness of the substrate and that the wear resistance was significantly improved. However, most of the studies have been focused on the preparation of coatings under one process, and there have been few performance comparisons between CLC alloy coatings and UHSLC alloy coatings. Therefore, it is necessary to investigate the differences in the microstructure and wear resistance properties of CLC alloy coatings and UHSLC alloy coatings.

During this period, a number of scholars have compared the differential performance of CLC and UHSLC coatings. For example, Zhang et al. [12,13] prepared CoCrFeMnNi high entropy alloy (HEA) coatings and CoCrFeNi coatings via high-speed laser cladding and normal laser cladding processes, respectively. The results show that laser cladding prepared coatings not only provide uniform and finer grain but also exhibit superior wear and corrosion resistance. Furthermore, CoCrFeMnNi HEA coatings are able to maintain good strength and toughness even in harsh environments. Yuan et al. [14] were deposited the Ni45 powders on a steel substrate with traditional low speed laser cladding and high-speed laser cladding processes, respectively. The results show that as the cladding speed increased, the wear and corrosion resistance of the cladded coatings became better. However, the preparation of UHSLC coatings and CLC coatings has not been more closely linked to practical engineering applications.

In this study, the train wheel material was used as the substrate; Fe-based alloy coating and Co-based alloy coating were prepared on the surface of wheel materials using the CLC process and UHSLC process; and the microstructure, element distribution, phase composition and wear morphology of CLC alloy coatings and UHSLC alloy coatings were compared and analysed. Furthermore, the tests were used to simulate the operation of high-speed trains and were closely linked to actual engineering applications to provide

theoretical guidance on the engineering applications of laser cladding for repairing wheel damage and surface modification manufacturing. At present, Fe-based and Co-based self-fusing alloy powders are widely used in wheel and rail materials. It is mainly due to the simple preparation and low cost of self-fusing alloy powders, as well as the excellent overall performance of Fe-based alloy coating and Co-based alloy coating [15,16].

## 2. Materials and Methods

### 2.1. Material and Specimen Preparation

Specimens were taken from the wheel and rail of an actual service rolling stock. The wheel specimens were taken from the area approximately 5 mm below the tread of the ER8 wheel, while the rail specimens were obtained from the top of the U71Mn hot-rolled rail. The sampling locations and base material dimensions are shown in Figure 1. The wheel and rail materials and alloy powder chemical composition are shown in Tables 1 and 2.

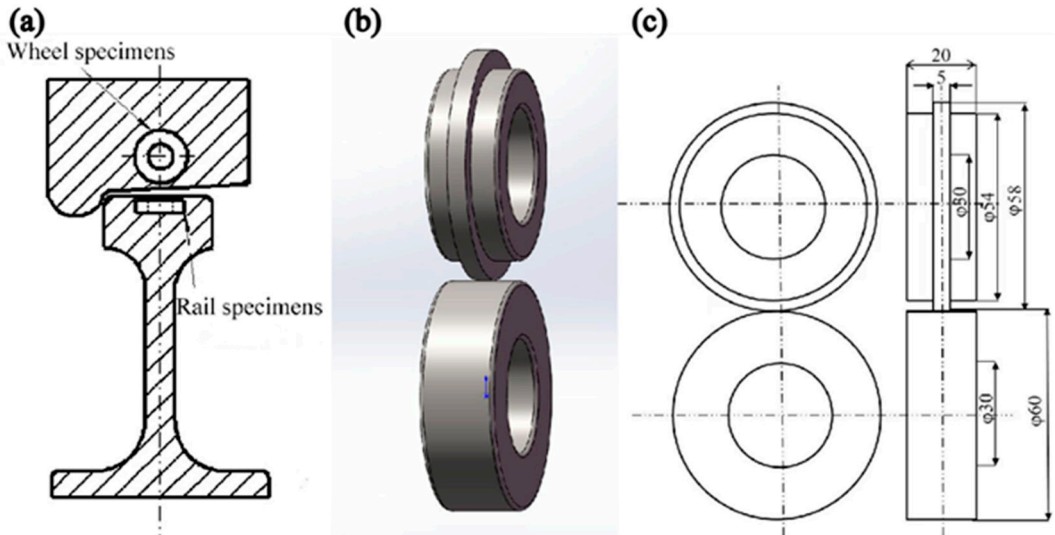

**Figure 1.** Diagram of the specimens. (**a**) Sampling position of the wheel/rail specimens. (**b**) Schematic diagram of the wheel/rail specimens. (**c**) Size of the wheel/rail specimens.

**Table 1.** Chemical compositions of the wheel/rail materials (wt%).

| Material | Element | | | | | | |
|---|---|---|---|---|---|---|---|
| | C | Si | Mn | P | S | Cr | Fe |
| RE8 wheel steel | 0.52–0.56 | 0.26–0.40 | 0.73–0.80 | ≤0.020 | ≤0.015 | 0.25–0.30 | Bal. |
| U71Mn hot-rolled steel | 0.71–0.80 | 0.50–0.80 | 0.70–1.05 | ≤0.030 | ≤0.030 | - | Bal. |

**Table 2.** Chemical compositions of the Fe-based alloy powder and Co-based alloy powder (wt%).

| Powders | Element | | | | | | | | |
|---|---|---|---|---|---|---|---|---|---|
| | C | Si | Mn | B | Cr | W | Ni | Fe | Co |
| Fe-based | 0.8–1.2 | 1.0–2.0 | 0.5–0.80 | 3.0–4.0 | 16.0–18.0 | - | 2.0 | Bal. | - |
| Co-based | 1.15 | 1.1 | - | - | 29.0 | 4.0 | ≤3.0 | ≤3.0 | Bal. |

Compared with the CLC process, from the energy distribution, the substrate absorbs more light energy than the powder particles in the CLC process, and the substrate absorbs energy to form a molten pool, which melts the powder delivered to the pool (Figure 2a). In contrast, the UHSLC process changes the energy distribution, and the powder particles

absorb more energy than the substrate. Therefore, the UHSLC process adjusts the convergence position of the laser, powder beam and molten pool so that the powder beam convergence point is located above the molten pool (Figure 2b). Then, the high-energy laser beam melts the powder material in the air, which condenses to the solid state on the substrate surface and forms a metallurgical bond with the substrate. Laserline LDF6000-100 (Shandong Mining Machinery Group, Weifang, China) and ZF-R6000-60 laser cladding systems (Jiangsu Zhufeng Technology Co., Ltd., Zhenjiang, China) were used to prepare CLC and UHSLC coatings on the ER8 wheel steel surface, respectively. The laser cladding process parameters are shown in Table 3.

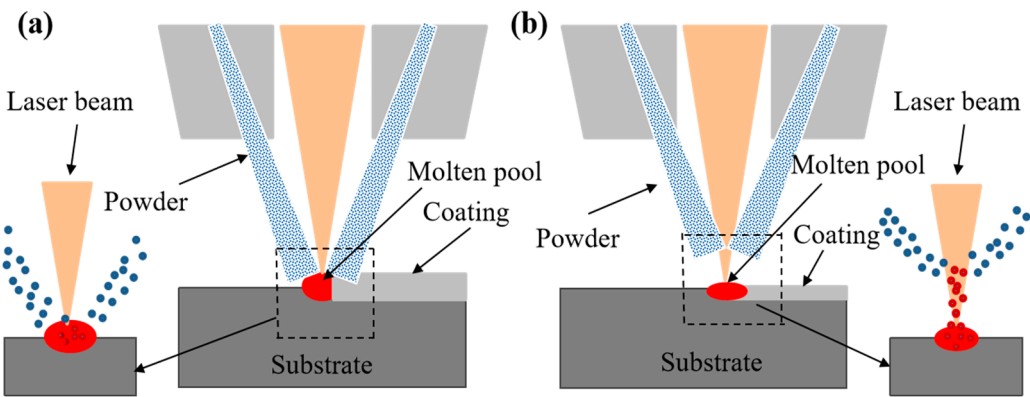

**Figure 2.** Schematic process principle of conventional laser cladding (CLC) and ultra-high-speed laser cladding (UHSLC). (**a**) CLC process. (**b**) UHSLC process.

**Table 3.** The parameters of the CLC process and UHSLC process.

| Laser Cladding Process | Parameter | | | | |
|---|---|---|---|---|---|
| | Power of Laser (W) | Scanning Speed (m/min) | Diameter of Light Spot (mm) | Powder Delivery Capacity (g/min) | Offset (mm) |
| CLC | 2200 | 0.4 | 3 | 21 | 1.5 |
| UHSLC | 2200 | 12 | 3 | 55 | 1.8 |

Figure 3 shows the preparation process of the test specimens, and the wheel specimens with a diameter of 58 mm were used as the substrate. Figure 3a shows the UHSLC process platform which consisted of three components: a high-speed lathe, a laser and powder feed system, and a wheel specimen. In this case, the powder feeding system and the laser were integrated in the laser head at the front of the control system, and the preset trajectory was set by means of the programmed control. During the laser cladding process, the machine drove the rotation of the pretreated wheel specimen. The laser melted the powder and a small amount of the substrate, forming a molten pool on the surface of the substrate. After removal of the laser beam, the self-cooling solidified to form a coating, and a uniform alloy coating covering the entire surface of the wheel specimen was obtained. Unlike the CLC process, the laser focus and powder focus in UHSLC were located above the surface of the wheel specimen, and the powder was heated and partially melted before it touched the surface of the wheel specimens. The coating with a thickness of 1 mm (Figure 3c) was processed on the surface of the untreated specimen using CLC and UHSLC systems. Finally, the wheel and rail specimens were both 60 mm in diameter. After preparation, all specimens were ultrasonically cleaned and dried.

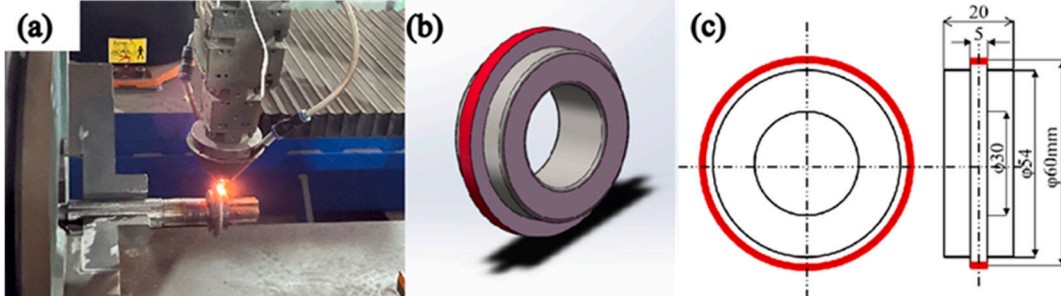

**Figure 3.** Laser cladding alloy coating specimen preparation process and coating specimen size. (**a**) Actual processing of the UHSLC process. (**b**) Diagram of alloy coating on wheel specimen. (**c**) The size of alloy coating on wheel specimen.

### 2.2. Test Parameters

In this paper, the test was completed with the help of a GPM-30 rolling contact fatigue testing machine (Jinan Yihua Tribology Technology Co., Ltd., Jinan, China) and Figure 4 shows the structure of the GPM-30 tester. The test simulated a rolling stock with an axle weight of 14 t and a running speed of 250 km/h class. The corresponding maximum contact stress $\sigma_{max}$ = 1100 MPa between the wheels and rails can be found according to Hertz theory [17].

$$N_{Simulation\ wheel} = \frac{\omega}{2\pi} \tag{1}$$

$$\omega = \frac{v}{R} \tag{2}$$

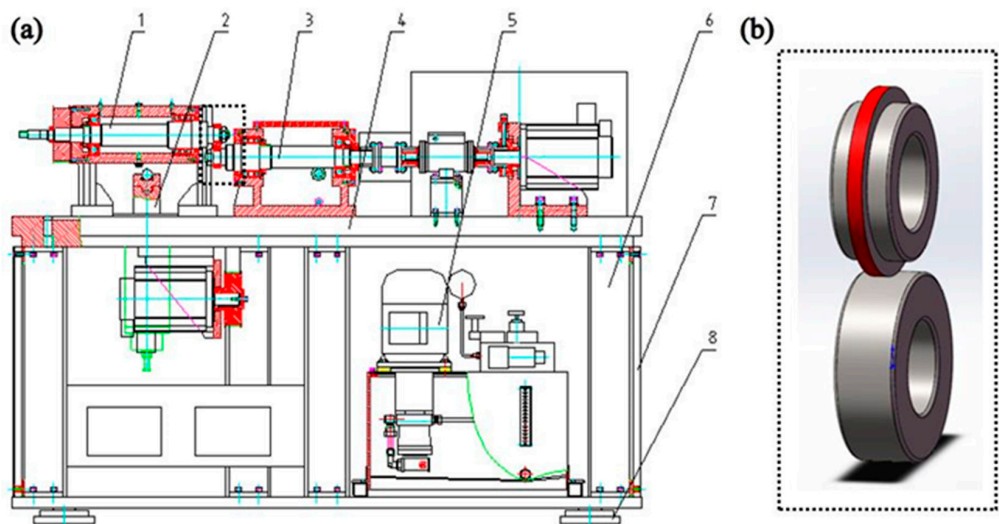

**Figure 4.** (**a**) Structure diagram of the main machine: 1—Accompanying test movement system, 2—Hydraulic actuator, 3—Active axis movement system, 4—Table, 5—Hydraulic pump station, 6—Frame, 7—Housing, 8—Adjusting pads. (**b**) Schematic diagram of wheel/rail specimens.

In Equations (1) and (2): $N_{simulation\ wheel}$—the rotational speed of the simulated wheel; $\omega$—the actual angular speed of the train wheel; $v$—the actual speed of the train wheel; and $R$—the actual radius of the train wheel. According to Equations (1) and (2), where the running speed v = 250 km/h and the wheel radius R = 430 mm (for the actual wheel radius), the test speed was found to be 1440 r/min. The slip rate was set to 0.4%, and the number of cycles was $1.8 \times 10^6$.

### 2.3. Microstructure Analysis and Properties of the Wear Resistance

After the cladding process was complete, the 10 mm × 10 mm × 5 mm block metallographic specimens were cut from the wheel specimens via wire cutting. The cross-sectional histomorphology and damage morphology of the coatings were characterized with the NEEEOHY optical microscope (OM, JE-68, , Beijing, China) and Hitachi scanning electron microscope (SEM, SU8010, Tokyo, Japan). The elemental composition and element distribution of the coatings were analysed with an energy-dispersive spectrometer (SU8010). The composition of the coating phases was studied with X-ray diffraction (XRD, CRD-6100, Tokyo, Japan). The scans were recorded between 10° and 90° with a with a scanning speed of 5°/min. The hardness of the coating was characterized with a Vickers hardness tester (Qness 10A+, Wien, Austria). Thus, the microstructure and wear resistance properties of the CLC and UHSLC alloy coatings were further analysed.

## 3. Results and Discussion

### 3.1. Microstructures

Figures 5 and 6 show the SEM micrographs and EDS results of the Fe-based alloy coatings and Co-based alloy coatings, respectively. The element distribution of both the CLC and UHSLC coatings was uniform, and the bonding area of the alloy coatings prepared using the CLC process was wavy, while the bonding area of the UHSLC coating wsa closer to a straight line. The coatings prepared using both laser cladding processes formed a good metallurgical bond, and no cracks or pores were found, indicating that they have good serviceability and can meet the requirements of various applications. By comparison, the thickness of the CLC coating was approximately 1000 μm (Figures 5a and 6a), and the thickness of the UHSLC coating was only 200 μm (Figures 5b and 6b), which is obviously much smaller than the former. This is mainly because the cladding speed of the UHSLC process is an order of magnitude faster than that of the CLC process, which results in a significant reduction in the coating thickness [18].

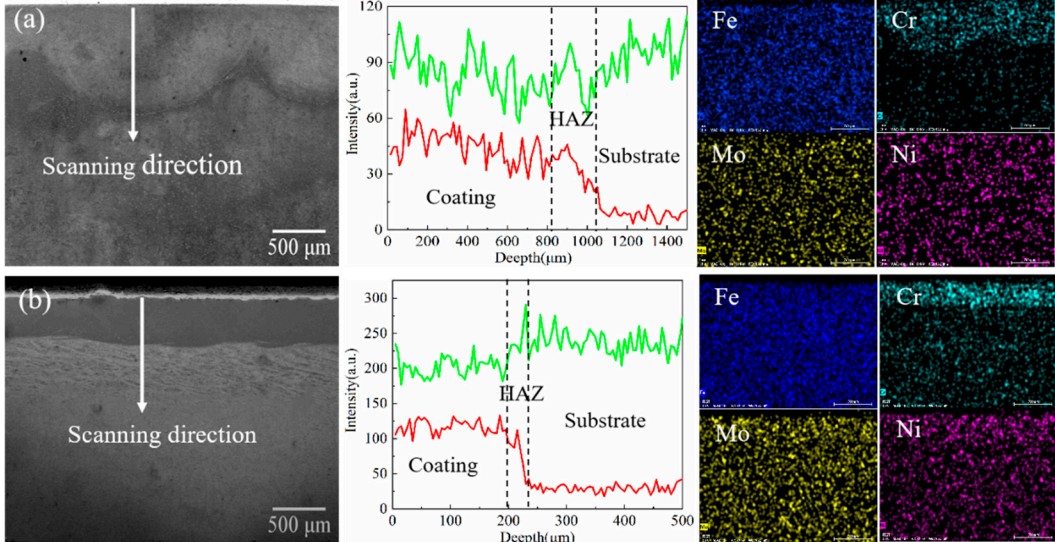

**Figure 5.** EDS line scanning results of elements in the transition zone and EDS surface scanning results of Fe-based alloy coatings. (**a**) CLC Fe-based alloy coating and (**b**) UHSLC Fe-based alloy coating.

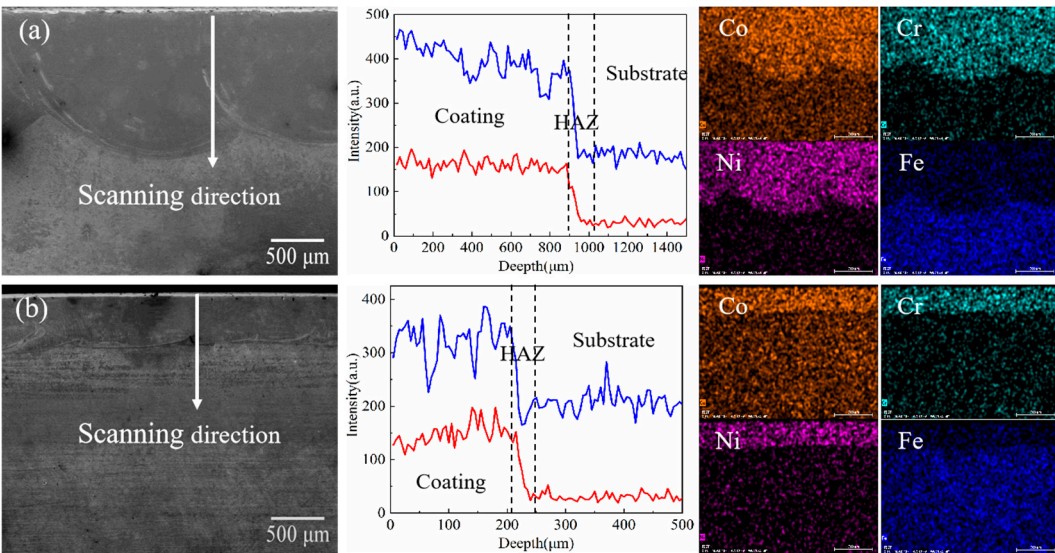

**Figure 6.** EDS line scanning results of elements in the transition zone and EDS surface scanning results of Co-based alloy coatings. (**a**) CLC Co-based alloy coating, (**b**) UHSLC Co-based alloy coating.

According to the EDS line scanning results, the element distribution of both the CLC and UHSLC coatings was relatively uniform, and the Cr element showed a nonclipped transition distribution on the surface of the coating and the substrate. The thickness of the heat-affected zone (HAZ) of the coating prepared with the same process was approximately equal. The CLC coating was approximately 150 μm (Figures 5a and 6a), and the HAZ of the UHSLC coating was approximately 40 μm (Figures 5a and 6a). This is because the UHSLC process has a much higher scanning speed than the CLC process with the same laser beam energy, which results in a low dilution ratio of the UHSLC process, therefore resulting in a much lower energy density on the substrate surface. The energy density obtained on the surface was greatly reduced. As a result, fewer elements diffused around the main body during the coating melting process, and the degree of segregation was low, effectively suppressing the detrimental effect of the coating composition deviating from the original powder composition due to substrate melting and solidification. At the same time, the thickness of the HAZ of the coating also decreased due to the low energy density of the substrate surface. In contrast, when alloy coatings are prepared with the CLC process, the substrate surface absorbs a large amount of energy generated by the light source. Therefore, it makes the energy density of the substrate surface larger and the depth of substrate melting greatly increases, thus forming a HAZ with a larger area, which also means that the CLC process has a higher dilution ratio [19].

Figure 7 shows the microstructures of the central region of the Fe-based and Co-based alloy coatings using the CLC process and UHSLC process. It is obvious from the figure that the microstructures of the four alloy coatings were composed of dendritic and reticulated eutectics. The growth direction of dendrites was disorganized and had a tendency to grow continuously to the outside. Compared with the CLC alloy coatings (Figure 7a,c), the UHSLC alloy coatings (Figure 7b,d) had a denser arrangement between the grains and a finer grain size. On the one hand, the UHSLC process scanning speed is very fast, and the energy density on the coating accumulates less, so the cooling speed of the coating melt pool in the melting process is increased accordingly. The grain growth process is limited by heat and time, which promotes the nucleation of the coating surface and inhibits the spreading growth of grains. Finally, the grain growth of the UHSLC coating is inhibited and shows the phenomenon of grain refinement [20,21].

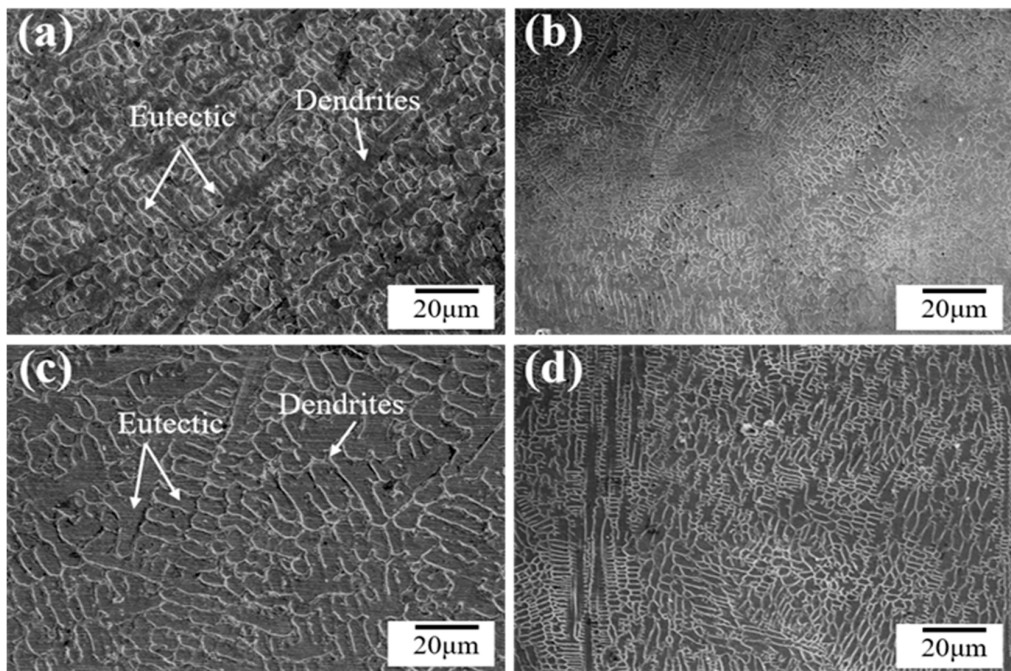

**Figure 7.** Microstructures of the specimens. (**a**) CLC Fe-based coating on the wheel specimen. (**b**) UHSLC Fe-based coating on the wheel specimen. (**c**) CLC Co-based coating on the wheel specimen. (**d**) UHSLC Co-based coating on the wheel specimen.

On the other hand, the change in solidification rate (R) further leads to a change in the shape control factor (G/R), which increases with the scanning speed, resulting in a smaller G/R and, therefore, more fine grains. Additionally, many studies have shown [22–24] that G/R has a significant effect on the structural phase of the coating organization.

Among them, the Fe-based coating dendrites were more robust, while the Co-based coating appeared to be finer, again due to the shape control factor (G/R) value that determines the microstructure of the alloy coating. Overall, the grains of the coatings prepared from the two alloy powder materials showed a dense distribution and no cracks [25,26].

Figure 8 shows the distribution of elements in the dendrites and eutectics of the coatings. The distribution of elements in the same laser cladding coating was not obvious. The distribution of elements in the dendritic and eutectic tissues was similar between the coatings. The main elements between the grains of the Fe-based alloy coating were Fe and Cr; the main elements between the grains of the Co-based alloy coating were Fe, Cr and Co. It is worth noting the distribution of Cr. Compared with the CLC alloy coatings, the UHSLC Fe-based and Co-based alloy coatings had higher Cr contents and lower elemental segregation.

Figure 9 shows the XRD results of Fe-based coatings and Co-based alloy coatings with different laser cladding processes. The Fe-based alloy coatings consisted of (Fe, Ni) and $Cr_7C_3$. This is because Fe-based alloy powder is composed of Fe, Ni and Cr elements, which are all third-period elements with similar atomic radii and can replace each other, so it is easy to form the (Fe, Ni) solid solution. The (Fe, Ni) solid solution leads to solid-solution strengthening of the coating and the formation of a more solid structure while also maintaining good toughness. At the same time, Cr and C elements in the alloy powder easily form carbide $Cr_7C_3$ under the action of faster condensation speed, and the formation of carbide effectively improves the hardness and strength of the cladding coating. The Co-based alloy coatings mainly consisted of the γ-Co phase and carbide $Cr_{23}C_6$, which is due to the rapid dissolution of Co in the alloy powder to form γ-Co under the action of a high-energy laser beam, and Co and Fe atoms have similar radii. The Fe atoms replace some of the Co atoms, forming a replacement solid solution. The Co and Fe elements

dissolve each other to inhibit their transformation to low-temperature structures, so the γ-Co solid solution is retained during rapid condensation [27].

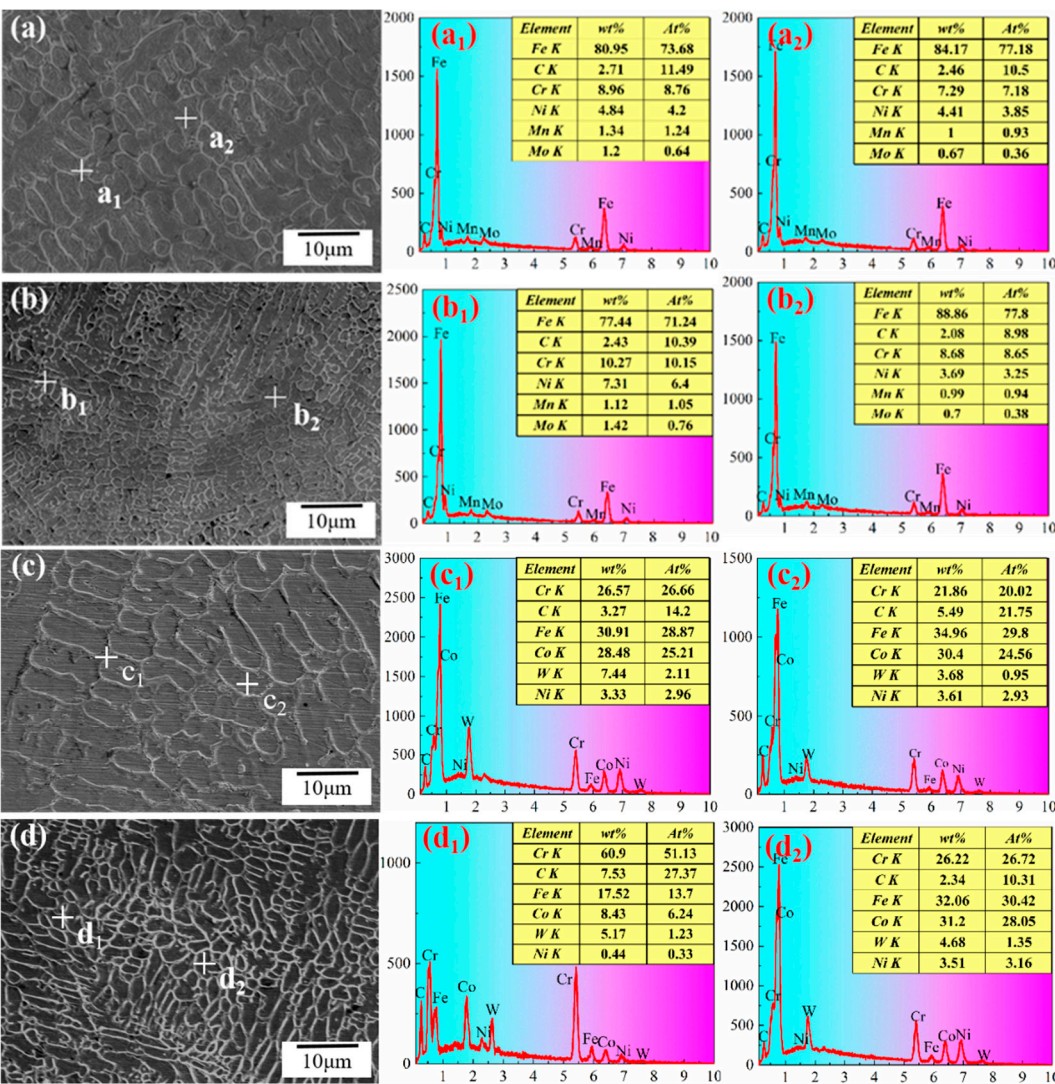

**Figure 8.** EDS point results of the Fe-based alloy coatings and Co-based alloy coatings at different laser cladding processes. (**a**) CLC Fe-based alloy coating: (**a₁**) Elemental content of dendrites; (**a₂**) Elemental content of eutectic; (**b**) CLC Co-based alloy coating: (**b₁**) Elemental content of dendrites; (**b₂**) Elemental content of eutectic; (**c**) UHSLC Fe-based alloy coating: (**c₁**) Elemental content of dendrites; (**c₂**) Elemental content of eutectic; (**d**) CLC Co-based alloy coating: (**d₁**) Elemental content of dendrites; (**d₂**) Elemental content of eutectic.

Hardness is an important parameter for evaluating material properties and depends mainly on the chemical composition and microstructure of the material. To better evaluate the wear properties of CLC coatings and UHSLC coatings, the hardness of each cladding coating cross section was measured using a Vickers hardness tester (Qness Q10A+, Austria) prior to conducting frictional wear tests. In this case, for the Fe-based alloy coating, a 1 N force was applied for 15 s. For the Co-based alloy coating, a 2 N force was applied for 10 s.

Figure 10 shows the hardness distribution of the four laser cladding alloy coatings. The hardness of all coatings was higher than the substrate hardness (303 HV) and decreased sharply at the HAZ, attributed to the low energy absorbed in the heat-affected zone of the coating, which is insufficient for phase transformation to occur. The HAZ of the UHSLC alloy coatings was approximately 58 μm, while the HAZ of the CLC coatings was larger

at 150 μm, indicating that the UHSLC process can effectively reduce the heat input to the substrate, thus improving the thermal stability of the coating. Compared to the hardness of the CLC coating, the UHSLC coating hardness was improved to a certain extent. Among them, the average hardness of the UHSLC Fe-based and Co-based alloy coatings was 709 HV and 525 HV, respectively. The average hardness of the CLC Fe-based and Co-based alloy coatings was 615 HV and 493 HV, respectively, which were improved by 15.3% and 6.5%. The main reason is that the UHSLC process generates more solid-solution phase, resulting in solid-solution strengthening, which leads to grain refinement [28]. In addition to the migration of C and Cr in the coating to the sides, the reaction forms hard carbides, which precipitate inside the grain and at the grain boundaries, and the combination causes the hardness of the UHSLC alloy coatings to increase [29].

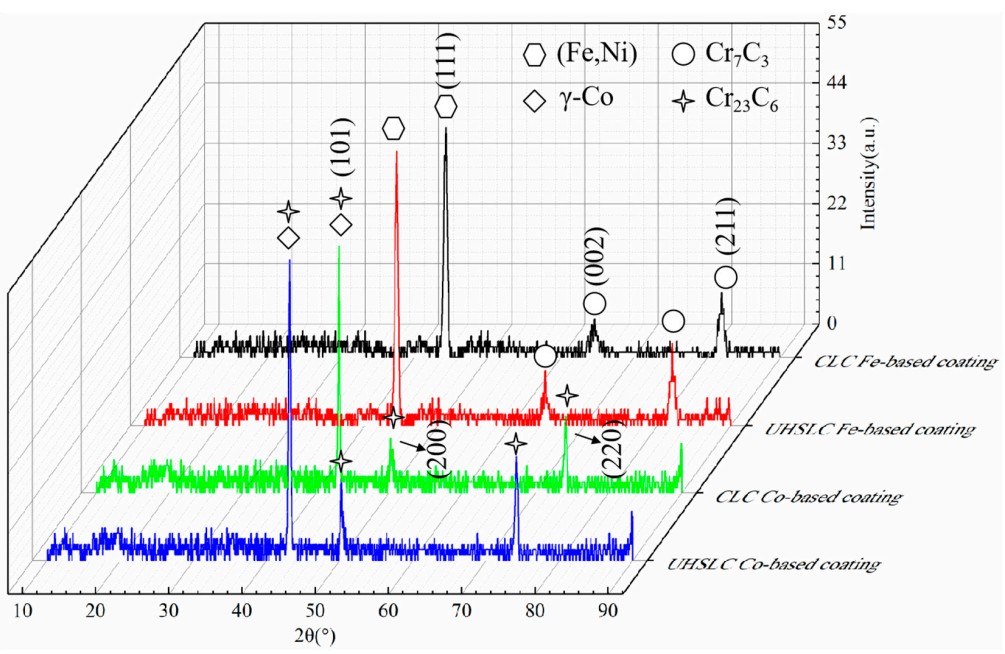

**Figure 9.** XRD results of Fe-based alloy coatings and Co-based alloy coatings with different processes.

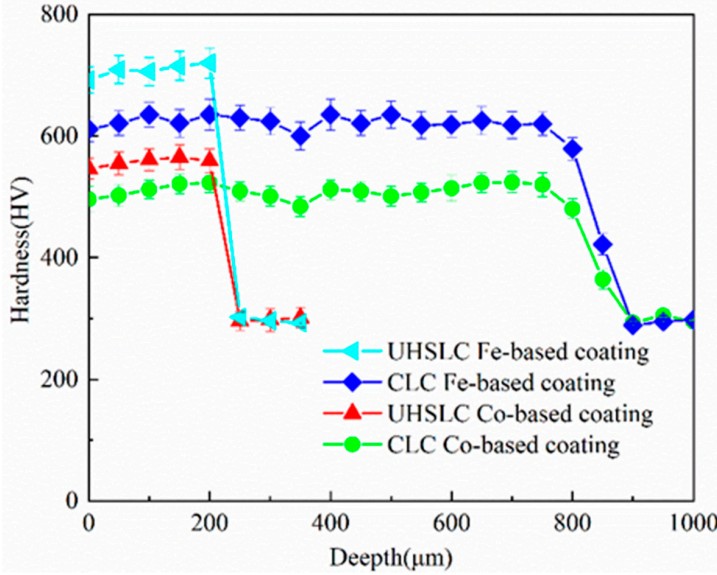

**Figure 10.** Hardness distribution of laser cladding coatings.

### 3.2. The Behaviour of Friction and Wear

Figure 11 shows the coefficient of friction (COF) curve and wear rate of the CLC and UHSLC alloy coatings on the wheel specimens tested for 180,000 cycles. The CLC Fe-based alloy coating on the wheel specimens had the largest COF (Figure 11a), and the UHSLC Co-based alloy coating on the wheel specimens had the smallest COF. This is due to the highest content of the element Fe in the Fe-based alloy powder, according to the principle of mutual solubility of the same metal materials; with increasing mutual solubility between metals, metal materials easily produce adhesion between each other, which leads to increased friction relations. The mutual solubility between different metal materials is low and cannot produce adhesion, so the COF is generally low. The Fe element content in the steel rail specimen is approximately 97%, and the Fe element content of the Fe-based alloy powder is higher than that of the Co-based alloy powder, so the COF curve shows the pattern shown in Figure 11a. The COF curve shows a clear trend and can be divided into a severe wear phase and a stable phase. The former is due to the smooth surface of the specimens and the existence of adsorption film before the start of the test, but with the rolling of the test, the adsorption film was destroyed, resulting in the roughness of the wheel surface, and the COF rose rapidly. After a period of running, the COF gradually stabilized and finally reached the kinetic equilibrium point. The CLC and UHSLC of the Fe-based and Co-based alloy coatings on the wheel specimens were stabilized at 0.39 and 0.31 and 0.35 and 0.28, respectively.

$$r = \frac{m_1 - m_2}{\pi d n} \tag{3}$$

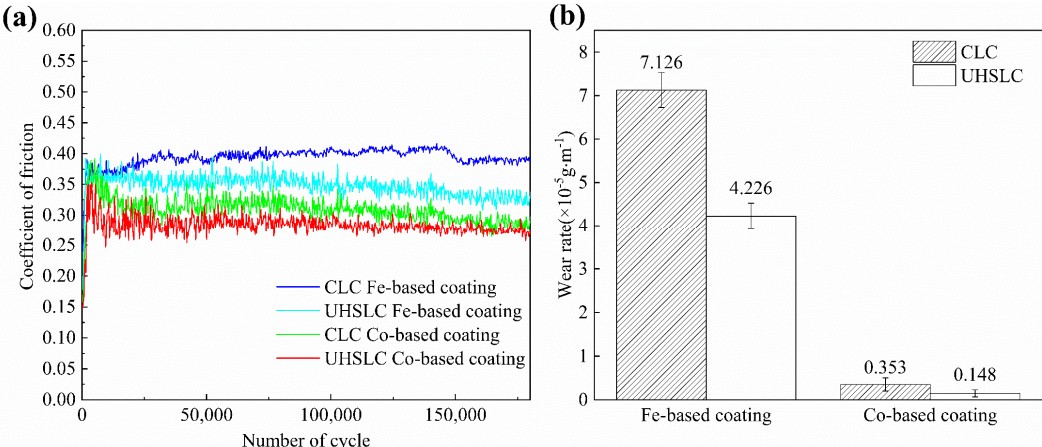

**Figure 11.** Coefficient of friction (COF) and wear rate of the wheel/rail specimens. (**a**) COF and (**b**) Wear rate.

In Equation (3): $r$—the wear rate of the wheel specimen; $m_1$—the mass of the wheel specimen before the experiment; $m_2$—the mass of the wheel specimen after the experiment; $d$—the diameter of the wheel specimen; and $n$—the number of test cycles. According to the calculation in Equation (3), the wear rate of the alloy coating can be calculated (Figure 11b). The highest wear rate of $7.126 \times 10^{-5}$ g·m$^{-1}$ was achieved for the CLC Fe-based coating on the wheel specimens, while the wear rate for the UHSLC Fe-based alloy coating on the wheel specimens decreased to $4.226 \times 10^{-5}$ g·m$^{-1}$, a reduction of approximately 40.7%. The wear rate of $0.148 \times 10^{-5}$ g·m$^{-1}$ for the UHSLC Co-based alloy coating was reduced by 58.1% compared to the CLC Co-based alloy coating.

Figure 12a,b show the surface damage profiles of the Fe-based alloy coatings on the wheel specimens. The surfaces of the wheel specimens all showed different degrees of damage. After a comparison, as shown in Figure 12a, a large number of spalling pits appeared on the grinding surface of the Fe-based coating on the wheel specimen with

the CLC process, accompanied by the occurrence of skinning. The SEM diagram clearly shows the sprouting of spalling pits and surface cracks. The form of damage was mainly fatigue damage, surface tissue peeling, and gradual expansion to the inner tissue. This is due to the repeated deformation of the surface tissue under the action of alternating stress during the rolling process of the specimens, which eventually reached the critical point of fatigue damage and flaked off from the material surface, forming spalling pits. In contrast, the surface of the UHSLC Fe-based alloy coating on the wheel specimens was smooth, as shown in Figure 12b, with slight damage and only a low degree of peeling, which indicate better wear performance [30]. This is mainly due to the rapid melting and solidification of Fe-based alloy coatings with the UHSLC process, which makes the microstructure fine-grained and strengthened. At the same time, the diffuse distribution of carbides inside the coating tissue can further strengthen the wear resistance of the coating and reduce the coating worn away by cutting using hard and equivalent substances. This is a key factor explaining why the wear rate of the UHSLC coating is much smaller than that of the CLC coating.

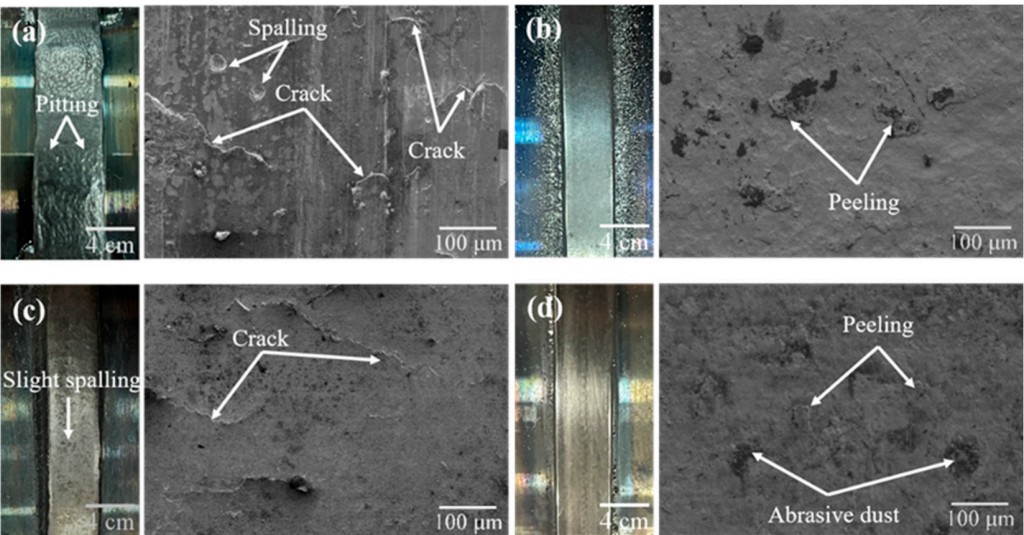

**Figure 12.** SEM micrographs showing the morphology of the worn surface of the specimens. (**a**) CLC Fe-based coating on the wheel specimen. (**b**) UHSLC Fe-based coating on the wheel specimen. (**c**) CLC Co-based coating on the wheel specimen. (**d**) UHSLC Co-based coating on the wheel specimen.

Figure 12c,d show the surface damage profiles of the Co-based alloy coatings on the wheel specimens. The surface of the Co-based alloy coating on the wheel specimens prepared using both processes was relatively smooth, with the former showing relatively minor spalling and peeling and the latter finding only a small amount of abrasive chips adhering to the specimen surface. A comprehensive comparison of the surface damage behaviour of Fe-based and Co-based alloy coatings on the wheel specimens prepared via CLC and UHSLC reveals that UHSLC alloy coatings provide superior wear resistance and can effectively reduce damage to the wheel surface, thereby improving wheel life and reliability. The surface damage of the Co-based alloy coating is less than that of the Fe-based alloy coating.

### 3.3. Damage Behaviour

Figure 13 shows the plastic deformation of the Fe-based alloy coating and Co-based alloy coating on the wheel specimens. For the same number of cycles, the degree and depth of plastic deformation of the wheel specimens with the CLC Fe-based alloy coating and Co-based alloy coating increased sequentially. The depths of the CLC Fe-based alloy coating and Co-based alloy coating were approximately 16 µm (Figure 13a) and 37 µm (Figure 13c), respectively. The degree of plastic deformation was mainly affected by the

size of the surface hardness, while the surface hardness of Fe-based alloy coatings and Co-based alloy coatings decreased in turn. The surface tissue was susceptible to slip in the direction of plastic flow due to tangential forces, thus making the wheel material plastic deformation degree larger. At the same time, the hardness of UHSLC coatings was generally higher than that of CLC coatings under the same material, the grain size of the coating tissue was significantly improved, and the strength was enhanced. Therefore, the depth of plastic deformation of Fe-based and Co-based alloy coatings via the UHSLC process was significantly reduced, and the depths were approximately 9 μm (Figure 13b) and 21 μm (Figure 13d), respectively.

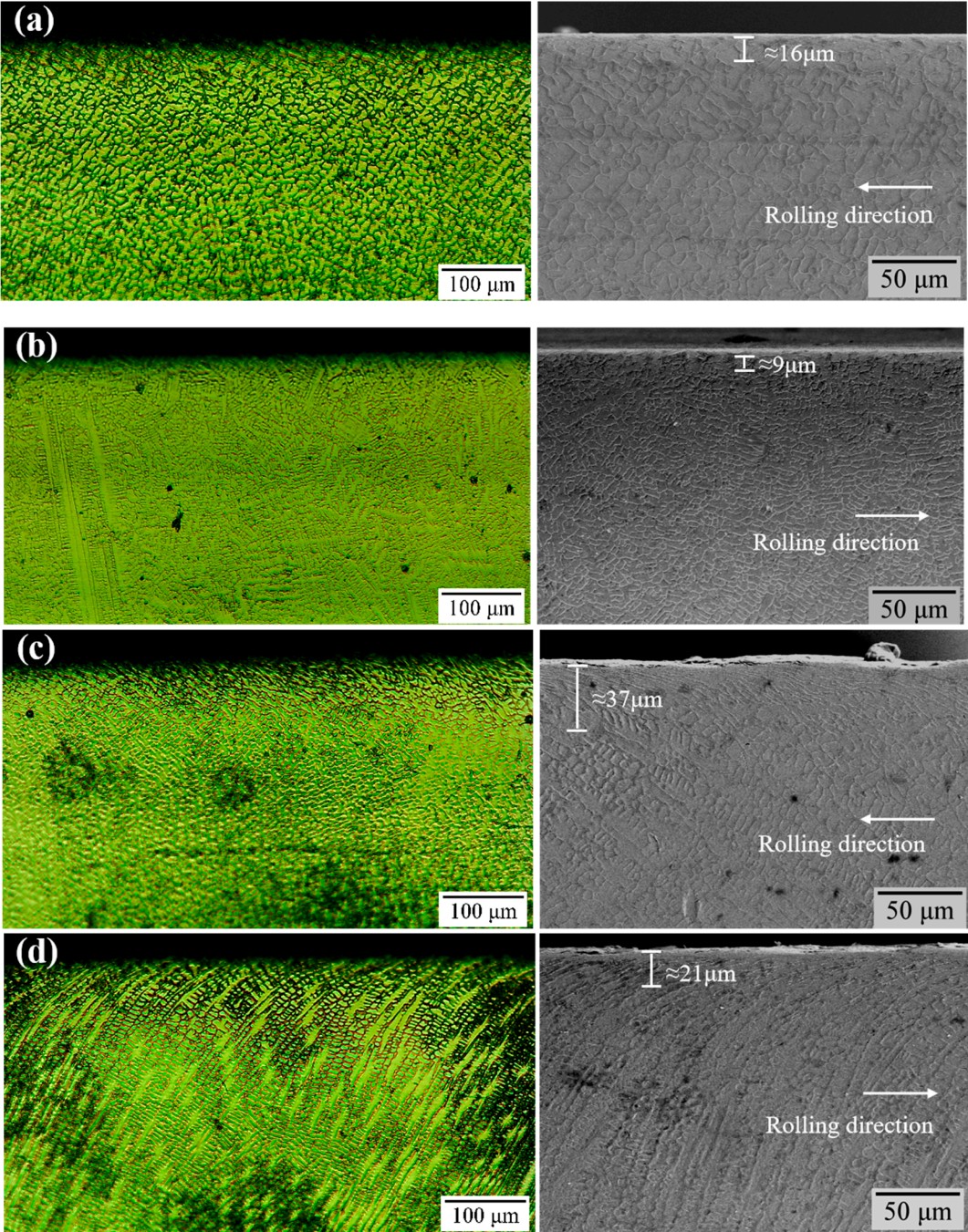

**Figure 13.** Plastic deformation diagram of wheel specimens. (**a**) CLC Fe-based alloy coating on the wheel specimen. (**b**) UHSLC Fe-based alloy coating on the wheel specimens. (**c**) CLC Co-based alloy coating on the wheel specimens. (**d**) UHSLC Co-based alloy coating on the wheel specimens.

Figure 14a,b show the SEM micrographs of the cross sections of the CLC Fe-based coating on the wheel specimens. Figure 14d,e show the SEM micrographs of the cross sections of the CLC Fe-based coating on the wheel specimens. After testing, it was found that the Co-based alloy coating on the wheel specimens showed significant fatigue cracks with a length of approximately 56 μm (Figure 14b) after treatment with the CLC process, and the cracks expanded to the inner tissue at a large angle through the crystal. Under contact stress, the Fe-based alloy coatings on the wheel specimens experienced severe friction and wear, leading to the development of cracks. When wheel specimens are subjected to tangential forces, significant slip occurs in the surface tissue, and this slip occurs in the direction of the tangential forces, thus affecting the wear properties and service life of the alloy coating [31]. At the same time, under the joint action of tangential and vertical forces, cracks sprouted and expanded along the direction of plastic deformation of grain tissue, and with the increasing number of rolling cycles, the fatigue cracks intensified until fracture. Finally, it peeled off from the surface of the wheel specimen [32], forming irregular peeling pits and leading to an increase in the wear rate of the Fe-based alloy coating on the wheel specimens. Compared with the Fe-based alloy coating, the degree of damage in the wheel specimen profile of the CLC coating Co-based alloy coating was significantly weakened. Minute cracks with a length of approximately 32 μm (Figure 14e) sprouted on the surface layer of the wheel specimens, and there was a tendency for the cracks to expand along the direction of tissue plastic deformation. On the one hand, this is because the Co-based alloy powder contains a large amount of Cr, thus contributing to the toughness of the coating. On the other hand, this is due to the UHSLC Co-based alloy coating on the wheel specimens having a smaller COF (Figure 11a). This leads to a reduction in the tangential force acting on the wheel specimens during the test, which to a certain extent inhibits the sprouting of cracks and, thus, substantially enhances the fatigue resistance of the wheel material.

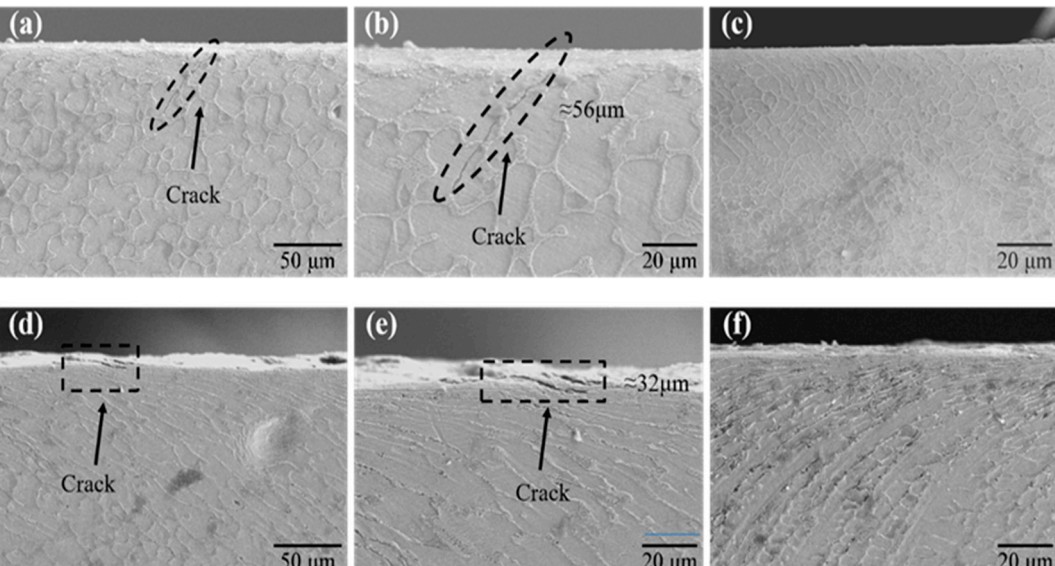

**Figure 14.** SEM micrographs of the cross sections of wheel specimens. (**a**,**b**) CLC Fe-based alloy coatings. (**c**) UHSLC Fe-based alloy coating. (**d**,**e**) CLC Co-based alloy coatings. (**f**) UHSLC Co-based alloy coating.

Figure 14c,f show the SEM micrographs of the cross sections of the UHSLC alloy coatings on the wheel specimens. It can be seen that no obvious fatigue cracks were found on the surface of both alloy coatings. This is because the hardness of the UHSLC coating is significantly higher than that of the conventional laser clad coating, and the COF is generally lower than that of the CLC coating, resulting in the UHSLC coating being subjected to a smaller tangential force while improving the toughness to a certain extent. In

addition, the fast melting and solidification characteristics of the UHSLC process strengthen the microstructures of the alloy coatings, laying a good foundation for the improvement in the coating strength.

## 4. Conclusions

(1) Under the specified process parameters, the grain structure of the Fe-based and Co-based alloy coatings prepared using the CLC and UHSLC process is tight, and there are no obvious defects, such as cracks and pores. The UHSLC process can significantly reduce the thickness of the Fe-based and Co-based alloy coating; the thickness of the CLC Fe-based and Co-based alloy coating is approximately 1000 μm, while the thickness of the UHSLC alloy coating is approximately 200 μm. Compared to CLC Fe-based and Co-based alloy coatings, UHSLC alloy coatings have a smaller heat-affected zone (HAZ), flatter coating–substrate bond, and smaller grain size. Fe-based alloy coatings mainly consist of a solid-solution phase (Fe, Ni) and carbide $Cr_7C_3$, and Co-based alloy coatings mainly consist of a $\gamma$-Co phase and carbide $Cr_{23}C_6$.

(2) The UHSLC process improves the hardness of the Fe-based and Co-based alloy coatings. Compared with the hardness of the CLC Fe-based and Co-based alloy coatings, the hardness of the UHSLC Fe-based and Co-based alloy coatings increased by 15.3% and 6.5%, respectively. The UHSLC process will produce more solid-solution phase, forming solid-solution strengthening, resulting in grain refinement, and thus enhancing the hardness of the UHSLC alloy coating. Among them, the hardness of the Fe-based alloy coating is greater than that of the Co-based alloy coating under the same laser cladding process.

(3) Compared with the CLC Fe-based and Co-based alloy coatings, the wear rate of the UHSLC Fe-based and Co-based alloy coatings was significantly reduced. The wear rate of the Fe-based alloy coating on the wheel specimens by the UHSLC process was reduced by 40.7%, and the wear rate of the Co-based alloy coating on the wheel specimens by UHSLC was reduced by 73.8%.

(4) The CLC Fe-based and Co-based alloy coatings on the wheel specimens showed more severe fatigue damage, while the UHSLC Fe-based and Co-based alloy coatings showed less damage and slight spalling. This indicates that the wear resistance properties of the UHSLC Fe-based and Co-based alloy coating were significantly better than those of the CLC Fe-based and Co-based alloy coating. Among them, the Co-based alloy coating showed better wear resistance properties under the same laser cladding process.

**Author Contributions:** Conceptualization, Q.X. and W.Y.; Methodology, Q.X., W.Y. and X.G.; Software, J.X. and X.G.; Validation, D.C., H.D. and W.Y.; Formal Analysis, J.X. and X.G.; Investigation, D.C. and Y.W.; Resources, J.X. and X.G.; Data Curation, X.G.; Writing—Original Draft Preparation, J.X. and X.G.; Writing—Review and Editing, Q.X., J.X. and Y.W.; Visualization, J.X. and D.C; Supervision, Q.X. and Y.W.; Project Administration, Q.X.; Funding Acquisition, Q.X., H.D. and D.C. All authors have read and agreed to the published version of the manuscript.

**Funding:** This research was funded by the National Natural Science Foundation of China (52202468); State Key Laboratory of Performance Monitoring and Protecting of Rail Transit Infrastructure (T2021102); Open Project of Key Laboratory of Conveyance Equipment (East China Jiaotong University), Ministry of Education (No. KLCE2021-10); and Research on key Technologies for laser cladding Repair of Railway Wheels and Rails (GJJ210665).

**Institutional Review Board Statement:** Not applicable.

**Informed Consent Statement:** Not applicable.

**Data Availability Statement:** The data that support the findings of this study are available from the corresponding author upon reasonable request.

**Conflicts of Interest:** The authors declare no conflict of interest.

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
