# Peer review of "Investigation of the Microstructure and Wear Properties of Conventional Laser Cladding and Ultra-High-Speed Laser Cladding Alloy Coatings for Wheel Materials"

_coatings, doi:10.3390/coatings13050949_

Round 1
Reviewer 1 Report
The article is devoted to a very topical topic, namely the study of laser cladding as an operation for the restoration of worn wheels of high-speed trains. The use of high-speed surfacing of wear-resistant compositions can significantly improve the operational properties of coatings of sheer parts. The article has scientific novelty and practical significance. During the research, modern equipment and methods were used. The article is well structured, contains high-quality graphic material in the form of photo microstructures, drawings of explaining the features of the ongoing studies.
During the research, the authors found that
1. The UHSLC process improves the hardness of the alloy coatings. Compared with the hardness of the CLC Fe-based alloy coating and Co-based alloy coating, the hardness of the UHSLC Fe-based alloy coating and Co-based alloy coating increased by 15.3% and 6.5%, respectively.
2. The thickness of the UHSLC alloy coating is approximately 200 µm.
3. The wear rate of the Fe-based alloy coating on the wheel specimens by the UHSLC process was reduced by 40.7%, and the wear rate of the Co-based alloy coating on the wheel specimens by UHSLC was reduced by 73.8%.
The results obtained are not in doubt. The article is written professionally and competently. I do not consider it necessary to make any corrections and additions to it. The article can be published in this version.
Author Response
Dear reviewer,
We quite appreciate your favorite consideration and the reviewer’s insightful comments concerning our manuscript entitled “Investigation of the microstructure and wear properties of conventional laser cladding and ultra-high-speed laser cladding alloy coatings for wheel materials”.
Thank you very much for your recognition and support of our work!

Reviewer 2 Report
1- In the abstract, the important gain results must be informed more deeply.
2- The introduction must be contained the similar previous work and inform the different in that work in more details
3- The XRD must be contained the XRD card number, which represented the presence characteristic peaks
4- In EDX analysis, the error percentages required to inserted
5- In experimental work, the the laser ablation process information must be discussed with more informative
Author Response
Dear reviewer,
We quite appreciate your favorite consideration and the reviewer’s insightful comments concerning our manuscript entitled “Investigation of the microstructure and wear properties of conventional laser cladding and ultra-high-speed laser cladding alloy coatings for wheel materials”. Those comments are helpful for revising and improving our paper, as well as the important guiding significance to our research. We have studied the comments carefully and have made a correction which we hope this revision can make our paper more acceptable.
Thank you very much for your time and consideration!
The main corrections in the paper and the responses to the reviewer’s comments are addressed point by point as follows:
Point 1: In the abstract, the important gain results must be informed more deeply.
Response 1: We are grateful to the reviewers for asking in-depth enough questions about the abstract section, in which we have described the phase composition of Fe-based and Co-based coatings, as well as comparing the differences in hardness between CLC and UHSLC alloy coatings. Please see follows.
The Fe-based coating was mainly composed of (Fe, Ni) and Cr7C3, and the Co-based coating was mainly composed of γ-Co and Cr23C6. It is found that the cooling rate of the CLC alloy coating was smaller than that of the USHLC, and the hardness of CLC alloy coating was less than that of the USHLC. The average hardness of the UHSLC Fe-based and Co-based alloy coatings are 709 HV and 525 HV, respectively. The average hardness of the CLC Fe-based and Co-based alloy coatings are 615 HV and 493 HV, respectively.
Point 2: The introduction must be contained the similar previous work and inform the different in that work in more details.
Response 2: We are very grateful to the reviewer for raising the issue of our inadequate introduction, in which we have cited references to similar previous work and have informed the different in our work. Please see follows.
During this period, a number of scholars have compared the differential performance of CLC and UHSLC coatings. For example, Zhang et al. [12,13] prepared CoCrFeMnNi high entropy (HEA) alloy coatings and CoCrFeNi coatings via high-speed laser cladding and normal laser cladding processes, respectively. The results show that laser cladding prepared coatings not only provide uniform and finer grain but also exhibit superior wear and corrosion resistance. Furthermore, CoCrFeMnNi HEA coatings are able to maintain good strength and toughness even in harsh environments. Yuan et al. [14] were deposited the Ni45 powders on steel substrate by traditional low speed laser cladding and high-speed laser cladding process, respectively. The results show that as the cladding speed increased, the wear and corrosion resistance of the cladded coatings became better. However, the preparation of UHSLC coatings and CLC coatings have not been more closely linked to practical engineering applications.
In this study, the train wheel material is used as the substrate, Fe-based alloy coating and Co-based alloy coating were prepared on the surface of wheel materials by CLC pro-cess and UHSLC process, the microstructure, element distribution, phase composition and wear morphology of CLC alloy coatings and UHSLC alloy coatings were compared and analysed. Furthermore the tests are used to simulate the operation of high-speed trains and are closely linked to actual engineering applications. To provide theoretical guidance on the engineering applications of laser cladding for re-pair wheel damage and surface modification manufacturing.
Point 3: The XRD must be contained the XRD card number, which represented the presence characteristic peaks.
Response 3: We are sorry for the confusion brought to the reviewer, as the rigour of the diagramming was not sufficient (Figure 9). Therefore, the XRD card numbers of the characteristic peaks were obtained with the help of the search in the JADE software and the comparison of PDF cards. And we have redrawn the Figure 9.
Point 4: In EDX analysis, the error percentages required to inserted.
Response 4: Thanks for your rigorous and professional comments. We have reviewed extensive literatures and found the percentages of these elements to be accurate values. In addition, we have carried out multiple scans of the same location of the coatings, the results are extremely repeatable and essentially error free. Your suggestions are very important for our research, and we will certainly take the helpful suggestions into consideration account in our subsequent work.
Point 5: In experimental work, the laser ablation process information must be discussed with more informative.
Response 5: Thank you for your professional comments, we have added more informative information to the text, in to hope that the reader can get a more detailed understanding of the laser cladding process. The details are as follows.
Figure 3(a) shows the UHSLC process platform which consists of three components: high-speed lathe, laser and powder feed system and wheel specimen. In this case, the powder feeding system and the laser are integrated in the laser head at the front of the control system, and the preset trajectory is set by means of the programmed control. Dur-ing the laser cladding process, the machine drives the rotation of the pre-treated wheel specimen. The laser melts the powder and a small amount of the substrate, forming a molten pool on the surface of the substrate. After removal of the laser beam, the self-cooling solidified to form a coating, and a uniform alloy coating covering the entire surface of the wheel specimen was obtained. Unlike CLC process, the laser focus and powder focus in UHSLC were located above the surface of the wheel specimen, and the powder was heated and partially melted before it touches the surface of the wheel speci-mens.
Thank you for your helpful suggestions and comments!

Reviewer 3 Report
There are other papers on the topic, but this is original. It is well referenced, but other references to similar work that should be included are https://doi.org/10.1016/j.jallcom.2023.169517
https://doi.org/10.1016/j.intermet.2022.107795
https://doi.org/10.1016/j.surfcoat.2020.126582
There is a mistake in Figure 2.The difference between conventional and UHSLC is in position of powder focal point not laser focal point. Table 3 shows diameter of the laser focal spot in both processes as equal but they are shown different in Figure 2.
Equations (1),(2): define symbols on first use.
Note: scanning speed is the input variable (under your control) and dilution is an the output variable (that results from the experiment), not the other way round (lines 170-173)
Make sure text in all Figures is big enough to read (Figures 6, 7, 11 and especially Figure 8, which a a long way from being readable)
Results and discussion and detailed and accurate. Many differences are due to faster cooling speed due to faster speed, but identifying the affect of differences (e.g. ‘ under the action of a high-energy laser beam, and the faster cooling speed of the material.
Figure 13: It is difficult to see any damage behaviour due to scale of the figures. If damage is to depth of a maximum of 37 um, and in some cases 5 um, why are micrographs approximately 300 x 200 um?
Conclusions are supported, but should note throughout that the conclusions apply to Co-based and Fe-based coatings under conditions ‘ ‘ rather than ‘CLC and UHSLC alloy coatings’ . This experiment provides useful evidence but cannot be generalised to all materials and all sets of parameters.
Author Response
Dear reviewer,
We quite appreciate your favorite consideration and the reviewer’s insightful comments concerning our manuscript entitled “Investigation of the microstructure and wear properties of conventional laser cladding and ultra-high-speed laser cladding alloy coatings for wheel materials”. Those comments are helpful for revising and improving our paper, as well as the important guiding significance to our research. We have studied the comments carefully and have made a correction which we hope this revision can make our paper more acceptable.
Thank you very much for your time and consideration!
The main corrections in the paper and the responses to the reviewer’s comments are addressed point by point as follows:
Point 1: There are other papers on the topic, but this is original. It is well referenced, but other references to similar work that should be included are https://doi.org/10.1016/j.jallcom.2023.169517
https://doi.org/10.1016/j.intermet.2022.107795
https://doi.org/10.1016/j.surfcoat.2020.126582
Response 1: Thank you for your helpful suggestions, we have read these articles carefully and cited them in our manuscript.
Point 2: There is a mistake in Figure 2.The difference between conventional and UHSLC is in position of powder focal point not laser focal point. Table 3 shows diameter of the laser focal spot in both processes as equal but they are shown different in Figure 2.
Response 2: We are very grateful to the reviewer for his professional comments., we are sorry for the confusion brought to the reviewer, as the rigour of the diagramming was not sufficient (Figure 2). So we have redrawn the Figure 2.
Point 3: Equations (1),(2): define symbols on first use.
Response 3:Thanks to the reviewer for the suggestions. We have defined the meaning of the symbols in the manuscript.
Point 4: Note: scanning speed is the input variable (under your control) and dilution is an the output variable (that results from the experiment), not the other way round (lines 170-173)
Response 4: We are very sorry for the expression errors in this manuscript. And thank you for your professional comments. We have made revisions in the manuscript. Please see follows.
This is due to the UHSLC process have a much higher scanning speed than the CLC process with the same laser beam energy, which results in low dilution ratio of the UHSLC process.
Point 5: Make sure text in all Figures is big enough to read (Figures 6, 7, 11 and especially Figure 8, which a long way from being readable)
Response 5: Thanks for your helpful suggestions, we have reformatted and resized the Figures in the manuscript to ensure that it has enough readable. After that, we will definitely pay more attention to the readability of the manuscript
Point 6: Results and discussion and detailed and accurate. Many differences are due to faster cooling speed due to faster speed, but identifying the affect of differences (e.g. ‘ under the action of a high-energy laser beam, and the faster cooling speed of the material.
Response 6: Thanks for your rigorous and professional comments. We are sorry for the misunderstanding brought to the reviewer. In the conclusions drawn in this paper, the phase composition of CLC and UHSLC alloy coatings are identical in the same podwer and we have discussed this section in more detail. The details are as follows.
Fe-based alloy coatings consisting of (Fe, Ni) and Cr7C3. This is because Fe-based alloy powder is composed of Fe, Ni and Cr elements, which are all third-period elements with similar atomic radii and can replace each other, so it is easy to form the (Fe, Ni) solid solution. The (Fe, Ni) solid solution leads to solid-solution strengthening of the coating and the formation of a more solid structure while also maintaining good toughness. At the same time, Cr and C elements in the alloy powder easily form carbide Cr7C3 under the action of faster condensation speed, and the formation of carbide effectively improves the hardness and strength of the cladding coating. The Co-based alloy coatings mainly consist of the γ-Co phase and carbide Cr23C6, which is due to the rapid dissolution of Co in the alloy powder to form γ-Co under the action of a high-energy laser beam, and Co and Fe atoms have similar radii. The Fe atoms replace some of the Co atoms, forming a replacement solid solution. And the Co and Fe elements dissolve each other to inhibit their transfor-mation to low-temperature structures, so the γ-Co solid solution is retained during rapid condensation.
Point 7: Figure 13: It is difficult to see any damage behaviour due to scale of the figures. If damage is to depth of a maximum of 37 um, and in some cases 5 um, why are micrographs approximately 300 x 200 um?
Response 7: Thanks for your helpful suggestions. In order to improve the readability of the images, we have re-sized the Figure 13. The OM and SEM images are sized to better compare the plastic deformed area with the undamaged area of the coaings.
Point 8: Conclusions are supported, but should note throughout that the conclusions apply to Co-based and Fe-based coatings under conditions ‘ ‘ rather than ‘CLC and UHSLC alloy coatings’ . This experiment provides useful evidence but cannot be generalised to all materials and all sets of parameters.
Response 8: We sincerely appreciate the valuable comments. We think that your suggestions are of great significance for this manuscript, and we have revised this section according to your comments.
Thank you for your helpful suggestions and comments!

Round 2
Reviewer 2 Report
The paper can be accepted in its current form
It was accepted